Trapping liquids may bias the results of beetle diversity assessment

Nakládal Oto 1
Havránková Eliška 2
Zumr Václav zumr@fld.czu.cz 1
1 Faculty of Forestry and Wood Sciences, Czech University of Life Sciences Prague , Prague , Czech Republic
2 Jovkova, Jordana Jovkova , Prague , Czech Republic
Huber Dezene
Electronic publication date: 2023 Dec 8
Publication date: 2023
Volume: 11
Electronic Location ID: e16531
Received 2023 Jul 1; Accepted 2023 Nov 6
Copyright: ©2023 Nakladal et al.
Copyright year: 2023
Copyright holder: Nakladal et al.
License: This is an open access article distributed under the terms of the Creative Commons Attribution License, which permits unrestricted use, distribution, reproduction and adaptation in any medium and for any purpose provided that it is properly attributed. For attribution, the original author(s), title, publication source (PeerJ) and either DOI or URL of the article must be cited.
License URL: https://creativecommons.org/licenses/by/4.0/

Keywords: Arthropoda, Coleoptera, Nitidulidae, Species richness, Bait trap, Saproxylic taxa

Funding: Internal Grant Agency of the Faculty of Forestry and Wood Science CZU A_01_22 This study was supported by the Internal Grant Agency of the Faculty of Forestry and Wood Science CZU (reg. no. A_01_22). The funders had no role in study design, data collection and analysis, decision to publish, or preparation of the manuscript.

==============================
Several different techniques and methods are used to capture and study beetles (Coleoptera). One option is the use of window traps with various trapping liquids. However, these liquids used in comparative studies may have a biasing effect on the results. The effectiveness of the frequently used liquid baits, involving beer, wine, vinegar, and water as the reference liquid, was compared in this study. Twenty-four traps were assigned to two habitat categories (sunny and shady) and four kinds of bait: beer, wine, vinegar, and water. During the study from June to July 2021, a total of 29,944 invertebrates were captured; of these, 3,931 individuals belonged to Coleoptera. A total of 3,825 beetles were identified, belonging to 120 species and 36 families. The most abundant family was Nitidulidae, with 3,297 adults (86% of the total). The number of arthropods differed only in the trapping liquid, and the captures were similar between beer and wine and between vinegar and water. The trapping liquid had a more significant effect on beetle abundance and species richness. In contrast, exposure had a significant effect only on the number of beetle species and a higher ratio of beetles was found in the shade. Beer and wine were very attractive and collected similar beetle communities. However, the diversity (Shannon’s index) was low due to the high abundance of several species. Traps with vinegar and water collected a similar composition and species richness. After removing sap beetles (Nitidulidae) from all traps, a significant difference was still recorded between trapping liquids in the number of individuals and species, and their communities were much more similar. Thus, at high abundances of sap beetles, it is possible to exclude them from analyses and obtain more accurate data when assessing environmental variables. The results showed that the type of trapping liquids used can have substantial effects on abundance and species composition captured beetles in traps especially for beer and wine. The beer and wine in traps can significantly influence the subsequent biodiversity assessment. We recommend the use of trapping liquids without the baiting effect to correctly assess the effect of environmental variables on beetle richness and abundance.

Introduction

Forest biodiversity has been frequent topic of study in recent decades (Oettel & Lapin, 2021), with beetles being a widely studied group (Seibold et al., 2015). Different methods of entomological data collection are used to assess species richness (Montgomery et al., 2021). They can be divided into quantitative and qualitative data acquisition, e.g., trapping, hatching (eclector traps) and sieving (Alinvi et al., 2007; Macagno et al., 2015; Vogel et al., 2021) or skimming of plant cover and attraction to light, Malaise traps, acoustic monitoring (Montgomery et al., 2021; Weiss et al., 2021), the last being manual collection (Mertlik, 2017). Trapping in pitfall and window traps is the most used method for beetles studies in scientific researches (Hohbein & Conway, 2018). The other methods of collecting entomological material are more suitable for collecting purposes and to observe life history, bionomy and phenology. Window traps are mostly used to assess the biodiversity of saproxylic beetles (Zumr, Remeš & Nakládal, 2022; Rothacher et al., 2023). Pitfall traps are used to assess different epigeic groups, especially beetles (Elek, Magura & Tóthmérész, 2001; Sroka & Finch, 2006; Podrázský, Remeš & Farkač, 2010) or spiders and ants (Černecká et al., 2020; Montgomery et al., 2021). Traps are largely not standardized especially in terms of size, colour and also the choice of liquids in traps (Brown & Matthews, 2016). In contrast to technical standardization, the choice of liquids is already determined by the objective of the study e.g., morphological, genomic and faunistical (Brown & Matthews, 2016). In contrast to pitfall traps, which have more often been the subject of comparative studies in terms of design and liquids used (e.g., Koivula et al., 2003; McCravy & Willand, 2007; Bell et al., 2014; Csaszar et al., 2018), the use of liquids in window traps is understudied. Each window trap has a reservoir with trapping liquid to capture and preserve entomological material for further analysis. Besides aqueous solutions with different preservatives, fermenting (attractive) liquids are also used to study beetle communities (Ruchin, Egorov & Khapugin, 2023). Some studies have used beer, wine, and vinegar as trapping fluids to assess beetle biodiversity in relation to surrounding environmental conditions (Ruchin & Egorov, 2021; Nakládal et al., 2022; Spina et al., 2023) or to monitor target groups, e.g., stag beetle (Lucanidae) (Bardiani et al., 2017), flower chafer (Scarabaeidae: Cetoniinae), longhorn beetle (Cerambycidae) (Touroult & Witté, 2020) or social wasps (Hymenoptera: Vespidae) (Dvořák & Landolt, 2006). The number of catches in traps is affected by the surrounding forest environment, e.g., dead wood (Seibold et al., 2015), successional stage of the forest (Hilmers et al., 2018), canopy openness (Lettenmaier et al., 2022) or trap placement on old trees (Parmain et al., 2018), among many others. However, the trapping liquid in traps may be variously attractive to arthropods. It thus may influence and bias the results of biodiversity assessments in relation to environmental variables, usually measured in the vicinity of the sampling units. Some species live in specific niches, such as Nitidulidae and many others, which inhabit and live in fermenting fruit or other rotting material and are more likely to be attracted by beer and wine. The research studies rarely pay attention to the effect of the trapping liquids used in the traps and focus only on the surrounding environmental characteristics studied. Large number of studies do not disclose the specification of the fluid used in the traps (Hohbein & Conway, 2018). For this reason, we tested the effect of various trapping liquids on beetle diversity assessment and under what conditions the use of individual trapping liquids for sampling beetles can be recommended to evaluate environmental variables.

Materials & Methods

Study site

The study was conducted in the protected area Nature Reserve (NR) Šance (49°58′N, 14°25′E) located in the territory of the capital city of Prague in the Czech Republic (Fig. 1). The reserve has an area of 198 ha with an altitude of 200–385 m above sea level. The site has been declared a protected site since 1982. The data collection permit has been approved by the nature conservation agency of the City of Prague (MHMP 644216/2021). The main subject of protection according to Šance (2022). (1) The native stands of arid acidophilous oak (dominant tree species Quercus petraea (Matt.) Liebl., Carpinus betulus L., Tilia cordata Mill.) forests rarely transitioning to areas of steppe character (relict scree forests on steep rocky slopes). Important forest plant associations (as. Cynancho-Quercetum, as. Melampyro nemorosi-Carpinetum, as. Aceri-Carpinetum). (2) Species of rare plants and animals mainly associated with scree forests. These oak woodlands grow on sunny slopes and slightly sloping plateaus with shallow, poor skeletal soil. In the upper parts of the slopes and on the ridges, they have the character of lowgrowing to old coppice stands of sessile oak. Cambisol is the predominant soil type. Lithosols and rankers are found on the slopes (Šance, 2022).

Figure 1 Maps of study location.

Location of study area of baiting study experiment (A–C). The design of each sampling unit grouped by different exposure to sunlight is indicated in the map (D). Map base created from Mapy.cz. (Creative Commons 4.0 (CC-BY-SA 4.0)).

Data collection and sampling design

Beetles were sampled during one month, June–July 2021. In total, 24 traps were installed on 22.06.2021 and uninstalled on 30.07.2021. Traps were installed 30 m apart in groups of four at six locations (three on sunny and three on shady locations, Fig. 1) by attaching the bucket with wire to the bole of various tree species at breast height (DBH 1.3 m). Traps in each group contained each one of four trapping liquid. The following mixtures were arranged in the traps: (i) red wine (mixture wine and sugar 1:1 and a spoonful of salt), (ii) beer (mixture beer and sugar 1:1 and a spoonful of salt), (iii) vinegar (8% acetic acid) and (iv) pure water as control. Sugar and salt are needed to start fermentation (a common practice with these liquids) (Touroult & Witté, 2020; Ruchin & Egorov, 2021). A total of 750 mL was added of the used trapping liquid mixture to all traps and the liquid was replenished after each trap collection. Control liquids received a drop of detergent to disrupt surface tension. Traps were picked every four days due to the rapid putrefaction of the trapped material. The trap consisted of a small bucket (diameter × height 13.2 × 11.8 cm, 1 litre volume) with a roof installed above (5 cm) to prevent rainfall and debris into the trap. The traps had no intercept windows or other tools for greater trapping effect (Fig. 2). The focus of the study was the order Coleoptera. This order was sorted into families and, in the next step, into the species level. Non-coleopteran taxa were quantified to the order level. Staphylinidae were determined to the species level except for the subfamily Aleocharinae, which was not determined because of its high difficulty. However, all individuals of Staphylinidae were counted. Taxonomic and other scientific nomenclature corresponds with Fauna Europaea (De Jong et al., 2014).

Figure 2 The trap types used in the baiting study.

Statistical analyses

We compared the effect of the trapping liquid and the trap light conditions (exposure) on the total number of beetles collected, species richness, and the abundance of individual species. Preliminary analyses showed that Nitidulidae comprised 86% of the beetles collected. Therefore, we first ran these analyses for all beetles. We then excluded Nitidulidae from the dataset and repeated these analyses.

The following analyses were performed in 4.3.1 (R Core Team, 2023). To compare differences between the trapping liquids and also exposure (explanatory variables) and the number of species and individuals per trap (fixed factor = response variables), a generalized linear mixed effect model with Poisson error distribution for species data and negative bionomical error distribution for abundance data was used. For this all-effects model (χ2), we used the package glmmTMB (Brooks et al., 2017). We used a nested design with random factors as different exposure for trapping liquid (1— exposure/hanging tree species) and for exposure fixed factor (1—trapping liquid/ hanging tree species). In these approaches with random effects, we wanted to find out more precisely the individual fixed factors in relation to other influential variables. The difference in the ratio of Coleoptera to all captured invertebrates was verified for non-integer values using the linear mixed effect model in package “nlme” with the function “lme” (Pinheiro & Bates, 2023). For multiple comparisons, differences between trapping liquids were analyzed in the package “emmeans” (Lenth, 2023) with post hoc Tukey HSD test and differences visualized by the package “multcompView” (Graves, Piepho & Dorai-Raj, 2023).

To evaluate Beta diversity, we used nonmetric multidimensional scaling NMDS (Bray-Curtis distance) to plot differences in communities within individual trapping liquids. The method creates an ordination space so that the distances among cases in this space best correspond to similarities or dissimilarities in its composition. For this approach, we used the package “vegan” with “metaMDS” function (Oksanen et al., 2022) to estimate the significant differences. The analysis of similarity (ANOSIM) method is used to evaluate a dissimilarity matrix (Clarke, 1993; Buttigieg & Ramette, 2014). All graphs were created in the package “ggplot2” (Wickham, 2016).

Subsequently, estimated species richness/diversity was generated using Inext software (Chao, Ma & Hsieh, 2016) based on the beetle abundance data. This rarefaction-extrapolation approach estimates the increase rate of species per number of individuals and was used to evaluate the attractive effect of each trapping liquid and exposure. Analyses were developed for general comparisons of the effect of individual trapping liquid versus different exposure. Number of bootstrap replications for compute was 50. This is a method for obtaining estimates of gamma diversity. Diversity indices are based on Chao et al. (2014) with Hill numbers: q = 0 (species richness) and q = 1 (exponential of the Shannon entropy index). Hill numbers are appropriate because they have distinct advantages over other diversity indices (Chao et al., 2014).

The following analyses were performed in CANOCO 5 software (Šmilauer & Lepš, 2014). Indicator species analysis (IndVal) was used to identify beetle species that indicate the attractivity of the individual trapping liquid (Dufrêne & Legendre, 1997). These results show the preference of recorded species to the specific habitat (Šmilauer & Lepš, 2014). From the abundance data, a minimum of four incidences and five individuals of species were included in the IndVal analyses.

Results

A total of 29,944 invertebrates were recorded, sorted in 16 orders. The five most abundant orders contained 99.5% of all invertebrates captured: Hymenoptera (17,624), Diptera (5,969), Coleoptera (3,931), Lepidoptera (1,759) and Blattodea (267). A different trapping effect was found between the trapping liquids on arthropods (Table 1). Beer and wine showed the highest attractiveness for arthropods, with a similar effect between exposure (Fig. 3). The trapping liquids did not show an increased ratio of target beetles (Coleoptera) from all the captured arthropods. On the contrary, the exposure showed a significant difference (Table 1). A higher ratio of non-target arthropods was recorded in beer and wine in sunny locations. Generally, a higher ratio of beetles (target group) was collected in shady locations (Fig. 3). Of the total recorded beetles, 3,825 were determined, belonging to 120 species and 36 families (Data S1). The most frequently collected family was Nitidulidae, with 3,297 individuals (99 non-Nitidulidae beetle species with 528 individuals). The most abundant species was Cryptarcha strigata (Nitidulidae), with 1,827 individuals, almost 48% of all the beetles. The most abundant beetle species are shown in Fig. 4.

Table 1 Results of generalized linear models (Poisson or negative binomial distribution) for the explanatory variables (trapping liquid and insolation) on the number of species and number of individuals of beetles (response variable).

Model parameters were tested using χ2 (F statistics were used for non-integer counts: Ratio). Significant effects are shown in bold (n.s. = non significant).

	Trapping liquid	Exposure	
All Arthropods	χ2(3)=99.97	p < 0.001	χ2(1) = 2.79	p= n.s.	
All beetle individuals	χ2(3)=18.57	p < 0.001	χ2(1) = 0.01	p= n.s.	
All beetle species	χ2(3)=35.86	p < 0.001	χ2(1)=4.65	p = 0.031	
exceptNitid species	χ2(3)=22.91	p < 0.001	χ2(1)=15.19	p < 0.001	
exceptNitid indiv.	χ2(3)=10.49	p = 0.015	χ2(1) = 2.17	p= n.s.	
Ratio	Df3 F1.82	p= n.s.	Df1 F 5.51	p = 0.037	

Figure 3 Boxplot of captured number of arthropods and ratio of beetle in arthropods.

Illustration of the total arthropods captured (A) and the ratio of the beetle abundance in the total arthropods captured by the trapping liquid (B) (beetle individuals/arthropod individuals = ratio %) divided by insolation. Solid lines indicate the median, the boxes indicate 25–75 percentile, and min-max values are error lines. The differences between trapping liquids according to insolation are indicated in the header of the graphs. Letters above bars indicate significance differences by multiple comparison post hoc Tukey test. (significance p value <0.001 ‘***’; p value <0.01 ‘**’; p value <0.05 ‘*’; n.s., non significant).

Figure 4 Overview of the most abundant beetle species.

The top ten species are shown. (Photographs taken with Leica and edited with CorelDRAW, 2021.5).

Beetles assessment

The overall number of species and abundance significantly differed between trapping liquids (Table 1). After removing the Nitidulidae, differences in species richness and abundance remained significant (Table 1). The effect of exposure was significant only for the species richness (Table 1). Number of beetle species and individuals per trap captured in trapping liquids is shown in Fig. 5. Wine and beer attracted a similar higher number of all beetle individuals and beetle species; a smaller number of captures were collected by vinegar and water. After removing the Nitidulidae, in relation to exposure, the abundance was not significantly different. However, the species richness showed a significant difference. More species were captured in sunny plots (Fig. 5). Species accumulation curves (Fig. 6A) indicate that wine has the highest species richness (number of species) and beer has the highest attraction in terms of total individuals collected, but after removing the most attracted family Nitidulidae, the differences disappeared (Fig. 6B). Shannon diversity showed the opposite trend, with water and vinegar having the highest species diversity (Fig. 6C). Nevertheless, after removal of Nitidulidae, these differences were not significant (Fig. 6D). The beetle communities significantly differ between the trapping liquids (NMDS, p < 0.001). Figure 7A demonstrates that beer and wine attract very similar beetle communities. After removing Nitidulidae, the clustering dissipated (p = 0.078, Fig. 7B), and the beetle communities were more similar. Individual trapping liquids had a higher (cumulative) number of species in sunny plots, in contrast to vinegar, which attracted more species and higher Shannon diversity in the shade (Data S1). Preference of beetle families and beetle species are shown (Table 2).

Figure 5 Box plot of captured number beetle species and beetle abundance.

Number of beetle all individuals (A) all species (B) and number individuals without Nitidulidae (C) and number of species without Nitidulidae (D) per trap between trapping liquids grouped by insolation. The median is indicated by the solid line in the box (25–75 percentile), and the error lines show min-max values. The differences in the trapping liquids in relation to insolation are indicated in the header of the graphs. The letters above bars indicate significance differences by multiple comparasion post hoc Tukey test. (significance p value <0.001 ‘***’; p value <0.01 ‘**’; p value <0.05 ‘*’ ‘; n.s., non significant).

Figure 6 Species cummulative curves (gamma diversity) of beetles captured in our study.

Sample-size-based rarefaction and extrapolation sampling gamma diversity curve showing Hill’s numbers (abundance data). (A, B) q = 0 (species richness) and (C, D) q = 1 (the exponential of Shannon’s entropy index). Colored shaded areas represent the 95% confidence intervals. Solid symbols represent a total number of species and extrapolation (dashed lines) up to double the reference sample size.

Figure 7 Overview of beetle communities captures in individual trapping liquids.

The nonmetric multidimensional metric scaling (NMDS) shows the similarity of samples based on the trapping liquids. All species (A) and excluded Nitidulidae family (B). One trap (beer) was excluded from the (B) graph due to only two trapped individuals that were not from Nitidulidae. The centroids of beetle communities are indicated by solid points.

Table 2 Preference of families and beetle species.

Characteristic families and beetle species (n, number of individuals) attracted by the individual trapping liquids. From the abundance data, a minimum of four incidences and five individuals of species were included for IndVal analyses (Dufrêne & Legendre, 1997)

Trapping liquid	Families	Species	
Beer	Buprestidae (n 12)
Lycidae (n 5)
Nitidulidae (n 3,297)
Oedemeridae (n 5)
Scraptiidae (n 12)
Staphylinidae (n 162)
Tenebrionidae (n 5)	Cryptarcha undata (n 845)
Epuraea melina (n 8)
Haptoncus ocularis (n 155)
Lygistopterus sanguineus (n 5)
Soronia grisea (n 67)
Xylotrechus antilope (n 15)	
Wine	Elateridae (n 69)
Carabidae (n 21)
Cerambycidae (n 42)
Scarabaeidae (n 52)	Melanotus crassicollis (n 57)
Stenurella melanura (n 10)
Xyleborus dryographus (n 18)	
Vinegar	Melyridae (n 12)	Dasytes plumbeus (n 10)	
Mordellidae (n 118)	
Water	Curculionidae (n 59)
Ptinidae (n 9)
Throscidae (n 13)	Aulonothroscus brevicollis (n 9)
Scolytus intricatus (n 12)		

Discussion

This study investigated that different trapping liquids can significantly impact the beetle richness and abundance evaluation. Several studies focused on the assessment of stand characteristics, e.g., volume and type of dead wood using liquid bait traps (beer, wine) (e.g., Redolfi De Zan et al., 2014; Ruchin & Egorov, 2021; Spina et al., 2023). Our findings show that especially the use of beer and wine is inappropriate for assessing the beetle species richness, as these trapping liquids gave results significantly biased from the reference liquid (water). For example, they also do not recommend the use of wine and vinegar in pitfall traps (Brown & Matthews, 2016). However, in our study vinegar was closest to the reference fluid with respect to captures of Arthropods and beetle species. This contrasts with Touroult & Witté (2020), who found liquid bait traps suitable for assessing species richness status. For example, with beer traps, Ruchin, Egorov & Khapugin (2023) captured a high number of saproxylic (deadwood dependent) beetles in meadow biotopes. Spina et al. (2023) searched significant correlations with some saproxylic species, and the use of beer traps can bias such results. In our study traps were located in identical environmental conditions; therefore, the trapping liquid used in the traps was of a major influence. Our traps did not have intercept windows, which, together with trapping liquids, could have a significant secondary trapping effect and, thus, a higher bias in the results. Window traps (without any bait) are a very effective and frequently used method of collecting entomological data (Okland, 1996; Alinvi et al., 2007). The most attractive trapping liquids in our study were beer and wine. These liquids contain ethanol, which is known as a very strong attractant for beetles (Bouget et al., 2009).

High numbers of other nontarget groups of invertebrates show that beer and wine can be recommended for regulation of dangerous insects or pests, e.g., (Rodríguez-Flores et al., 2019) or hymenopterans (Vespidae, Formicidae), which were the most attracted families in beer and wine also in our study. Moreover, the attractiveness of beer or wine can be used for monitoring target taxa, e.g., stag beetles (Bardiani et al., 2017) or overall effective monitoring of selected groups of invertebrates, e.g., social wasps (Dvořák & Landolt, 2006). Another suitable use for baited traps is monitoring the occurrence of species (e.g., nonnative new species), as the use of baits significantly increases the number of captured individuals and thus better chance detection, as in our case Haptoncus ocularis (Nitidulidae).

In our study, the trapping liquids can attract distinct beetle communities. This may be related to their life strategy, for example, the difference between saproxylic and non-saproxylic species (Bouget et al., 2009) or floricolous species (Ruchin, Egorov & Khapugin, 2021). The most frequently captured family was Nitidulidae, which is consistent with other studies using beer traps (Ruchin et al., 2021; Ruchin, Egorov & Khapugin, 2023). The most recorded species was Cryptarcha strigata, 48% of sampled beetles. This species feed on the fermenting sap of trees and their fruits, and their food generally contains small amounts of ethanol. Thus, traps with alcoholic liquids are highly attractive to C. strigata. Similar abundances (50%) of this species captured with beer traps have been reported by Ruchin, Egorov & Khapugin (2021).

Many studies excluded the family Staphylinidae due to the difficult determination (e.g., Parmain et al., 2015; Kozel et al., 2021). However, their exclusion does not bias the resulting assessment of beetle biodiversity (Parmain et al., 2015). In our study, the family Nitidulidae was the most attracted group, representing a significant percentage of abundance and species richness. A similar result was found by Ruchin & Egorov (2021) using beer traps. At the same time, similarly to Staphylinidae, these sap beetles are also very difficult to determine into species, especially the genera Epuraea.

After removing the family Nitidulidae, the results of the trapping liquid effect could be used to assess the species and abundance of beetles in specific environmental conditions. Even so, fermented liquids are attractive baits and will always partially bias the resulting data. The exposure effect was found to be significant only for the beetle species richness, although the significant difference between the trapping liquids was also recorded. A higher number of species was recorded in sunny plots, which implies that beetles prefer sunny habitats (e.g., Lettenmaier et al., 2022; Nakládal et al., 2022). Beer and wine consistently captured higher numbers of beetle species than vinegar and water in sunny plots. In contrast, vinegar attracted more beetle species (cumulative) in shaded stands. Water, as a reference liquid, showed the smallest differences in different exposure and, therefore, appears to be the most suitable trapping liquid.

However, insect captured in pure water undergoes strong decomposition; therefore, traps will need to be emptied more frequently or need to use some preservation matter. The trapping liquid (main component is water) are used with several different preservatives. Frequently used liquids are the following: brine (Sebek et al., 2016; Weiss et al., 2021), propylene glycol liquid (Joelsson, Hjältén & Work, 2018; Zumr et al., 2023), copper sulfate solution (Leidinger et al., 2021; Rothacher et al., 2023) or even the very toxic formaldehyde (Sroka & Finch, 2006). Further research could be aimed at verifying whether these liquids affect the trapping results.

Conclusions

In general, the use of an appropriate technique for collecting entomological data is an important parameter for a correct evaluation of the results. Our study verified the influence of the trapping liquid on beetle abundance and species richness. A significant difference from the reference liquid (water) was observed for beer and wine, as opposed to vinegar. Our findings showed that the trapping liquids can significantly bias the results of the assessment of environmental variables and their effect on beetle abundance and species richness. For faunistic research and to capture the maximum number of species, the use of baited traps is a suitable approach. Beer and wine are highly attractive for the family Nitidulidae, so it may be appropriate to exclude this family from the analyses and use the data to evaluate stand characteristics, even if not only one family is attracted specifically to these liquids. However, we recommend the use of trapping liquids without the baiting effect to assess the effect of environmental forest variables on beetle richness and abundance.

Supplemental Information

Data S1 Figures and abudance of captured beetles

Click here for additional data file.

Data S2 Raw data

Click here for additional data file.

We are grateful to the following experts for help in the identification of some beetle families: Jan Horák (Praha): Scraptiidae, Mordelidae; Pavel Průdek (Brno): Cerylonidae, Ciidae, Corylophagidae, Cryptophagidae, Latridiidae, Monotomidae; Josef Jelínek (Praha): Nitidulidae. We are also grateful the anonymous reviewers for their very high-quality comments to improve the manuscript.

Additional Information and Declarations

Competing Interests

Author Contributions

Field Study Permissions

Data Availability

The authors declare there are no competing interests.

Oto Nakládal conceived and designed the experiments, prepared figures and/or tables, and approved the final draft.

Eliška Havránková conceived and designed the experiments, performed the experiments, prepared figures and/or tables, and approved the final draft.

Václav Zumr analyzed the data, prepared figures and/or tables, authored or reviewed drafts of the article, and approved the final draft.

The following information was supplied relating to field study approvals (i.e., approving body and any reference numbers):

Nature Conservation Agency of Prague city for permission to conduct the research.

The following information was supplied regarding data availability:

The raw measurements are available in the Supplementary Files.

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
