# Peer review of "Trapping liquids may bias the results of beetle diversity assessment"

_PeerJ, doi:10.7717/peerj.16531_

## Round 0.1 · original submission · Major Revisions

Both reviewers have provided excellent and extensive comments on this manuscript. One point that came up in both reviews, and in several different ways, was the need to better define and state the purpose(s) and hypotheses of this study. This is particularly important due to the trapping context in which substances known to be attractant to some groups were used.

As this is a major revision, I expect to ask these and/or other reviewers to have a look at your response and revisions in a second round of review. (Note that there are also some language issues that should be improved.)

**Language Note:** The Academic Editor has identified that the English language must be improved. PeerJ can provide language editing services - please contact us at copyediting@peerj.com for pricing (be sure to provide your manuscript number and title). Alternatively, you should make your own arrangements to improve the language quality and provide details in your response letter. – PeerJ Staff

Reviewer 1 ·

Excellent Review

This review has been rated excellent by staff (in the top 15% of reviews)
EDITOR COMMENT
I would like to thank the reviewer for this exceptional review. The reviewer took a great deal of time with the manuscript and provided detailed and constructive comments to the authors that will help in improving the paper. I particularly appreciated the suggestions that the reviewer gave for better framing of the results.

Basic reporting

The authors compare recovery of insects (with a focus on beetles) in traps baited with different attractants (beer, wine, vinegar, water = control) located in sunny versus shaded areas. The type of attractant affects trap catch with similar catches between beer/wine versus vinegar/water. Most of the pattern is driven by recovery of large numbers of sap beetles (Nitidulidae) in beer/wine traps. This is consistent with information provided in the Introduction “… such as Nitidulidae, which inhabit and live in fermenting fruit or other rotting material and are therefore more likely to be attracted by beer and wine.” When sap beetles are removed from the dataset, trap catches are similar across the four types of attractants. The authors “recommend the use of unbaited traps to assess the effect of environmental forest variables on beetle richness and abundance”.

The table and figures are useful (but see comments below). The raw data is provided in a supplemental table. References are appropriate, but some of them could be removed (see #3 below). Additional text could be added to expand upon the findings. Extensive editing by someone fluent in the English language is required to correct or clarify wording in the main text, in the figure captions and in the figures.

Experimental design

The experiment is well-designed and the data supports the conclusions. Methods can be replicated with the information provided in the text.

Validity of the findings

Results of the analyses support the conclusions.

Additional comments

1) Abstract, main text, conclusions and throughout – refer to the solutions used as baits or attractants. Do not refer to them as preservatives or fixatives; i.e., from the abstract “… most commonly used fixative solutions (beer, wine, vinegar and water as reference liquid)”. None of these solutions have preservative properties as noted by the authors; e.g., “Traps were picked every four days due to the rapid putrefaction of the trapped material.” (lines 93-94).

2) A key aspect of the study is comparison of trap catch between sunny vs. shaded sites. Some mention of the results for this comparison should be made in the Abstract.

3) Introduction – Related to #1, consider removing discussion of preservatives in the Introduction. None of the solutions used were preservatives. Introducing the topic of preservatives is unnecessary and may confuse readers as to the focus of the current study.

4) Some examples of confusing wording
o Line 65 - “… we wanted to find out what effect the different attracting abilities of the baits may have and also to see how to leave this method in terms of frequent use in comparing different environmental niches.”
What does it mean to ‘leave this method’?
o Line 98 - “The other trapped order was only counted and identified to individual orders.”
Suggested rewording - “Non-coleopteran taxa were quantified to order.”
o Line 109 – “All methods were applied twice in the same appropriate way for all species (all beetles) and their abundances (all beetles) and numbers of species (without Nitidulidae) and their abundances (without Nitidulidae)”.
Suggested rewording - “We compared the effect of bait type on the total number of beetles recovered, species richness, and the abundance of individual species. Preliminary analyses showed that Nitidulidae comprised 86% of the beetles recovered. Therefore, we first ran these analyses for all beetles. We then excluded Nitidulidae from the dataset and repeated these analyses.”

5) Statistical analyses – Catches were combined across trap periods to obtain 1 sample/trap (n = 6 samples/bait type). Had the authors considered using as their samples, the number of beetles recovered in a trap for each 4-day trap period (ca. 7 samples/trap or ca. 28 samples/bait type)? I think I know why they didn’t, but it would be useful to the reader to specifically state why trap catches were combined across trap periods.

6) Lines 171, 173 – what is ‘pF’? (I’m not familiar with how degrees of freedom are reported in CCA and wonder if this is perhaps a typographical error.)

7) Suggestions to better frame results and conclusions in an appropriate context.
o Line 222 – “Water as a reference solution responded to exposure with the smallest differences and therefore appears to be the most suitable solution that does not bias anymore from the effect of exposure.” Consider also mentioning that, if only water is used (without addition of a preservative), traps will need to be emptied more frequently.
o Removing Nitidulidae from consideration leaves 634 beetles in the dataset or about 26 beetles/sample (< 1 non-Nitidulidae captured per trap/day). I agree that the authors have interpreted the results correctly, but caution that the findings are based on a somewhat small number of beetles. The authors may wish to include such a caution when presenting their findings.

8) Table 1 has an unusual format. Consider reformatting with three columns. To increase the value of the table, include in brackets, the number of individuals recovered for each taxa. The latter is provided in Supplemental Table 1S, but extracting this information for Families is more difficult (see #12)

Bait type Most common order (# inds.) Most common species (# inds.)
Beer Buprestidae (X), Cryptophagidae (X) Cryptarcha undata (X), Epuraea melina (X),
Wine
Vinegar
Water
(control)

9) The layout of the figures is confusing. Seventeen separate figures are identified, but I think some figures are intended as one figure with 2-panels. For example, there are separate captions for Figures 3 and 4 referring to illustrations on the left (Fig. 3) versus on the right (Fig. 4). If this is the case, only one caption should be provided for each 2-panel figure; the figures (e.g., 3 and 4) should be formatted as 1, 2-panel figure. Similarly, captions for Figures 10 and 11 suggest that they also are intended to be 1, 2-panel figure (and also 12 & 13, 14 & 15).

10) I encourage the authors to reevaluate the figures to see if some of them might be better suited as tables (e.g., Figs 3-9), or which figures might be better suited as supplementary material (e.g., Fig. 2)

11) RE: Figures 16 and 17
o in the body of Figure 16, vinegar is misspelled as ‘Vinegard’
o The figure captions need to direct the reader to text or a table that provides the full genus and species names related to the codes. For example, how is the reader supposed to know what is meant by ‘DromBarn’? For this information, the authors could refer the reader to Supplemental Table 1S.

12) RE: Supplemental file.
o The single figure in this file is labelled ‘Figure 9S’. It would seem more intuitive to label it as ‘Figure 1S’.
o To make Table 1S more useful to the reader, I suggest a new column be added to the left to list the species names by family. This would better allow the reader to identify how many beetles in each family were collected and aid comparison with Table 1. Genus/species codes can be provided in the 2nd column (refer to #11).

Family
Species Code Vinegar Beer Wine Water Sum
Buprestidae 5
Agrilus angustulus AgriAngu 1 1 2
Agrilus hastulifer AgriHast 2 2
Agrilus obscuricollis AgriObsc 1 1

Reviewer 2 ·

Excellent Review

This review has been rated excellent by staff (in the top 15% of reviews)
EDITOR COMMENT
I very much appreciate the extensive review provided by this referee. The reviewer provided substantial ideas and examples of how the authors could improve the manuscript – including ideas about better contextualization of the results and discussion of the statistical methodology. This review will be helpful in improving the manuscript.

Basic reporting

This is an interesting descriptive study comparing the effectiveness/efficiency of using different preservative solutions in trapping devices meant for collecting invertebrates (but focused on Coleoptera) in the Czech Republic.

The manuscript is relatively well written; however, as English is not the native language of the authors, there is considerable and significant room for improvement (throughout but particularly the statistical analyses and results sections). There are many instances in which statements are not very clear due mostly to the use of English, for which I provided some suggestions, although in some cases I was not very sure if my interpretation was what authors meant. At some point I stopped providing suggestions as I felt I was rewriting the manuscript for the authors. Thus, I strongly suggest the authors to read-proof their manuscript. Also, I think authors have largely restricted their literature review to European papers; thus, I suggest them to expand to the North American published sources.

Although I think I understand what authors mean with the title, I don’t think it is grammatically correct the way it is written. Perhaps ‘Different solutions in traps may bias…’ may be more appropriate. I don’t really get the ‘monitoring traps’ piece as well as ‘niche beetle diversity’. This comment is in relation to my previous one. Further, I really see no connection between the title (if I understand it correctly) with what is contained in the abstract and partially in the main text, as there is little linkage between what is provided with biases in the evaluation of beetle niche (it may be just the way the title is written) and partially with beetle diversity.

One major comment I have is the choice of solutions in the traps. If the focus of the study is to evaluate potential biases as a result of the solution used, a wider array of solutions should have been used. From the start, the study was already biased (intentionally or unintentionally), given the choice of solutions. Beer, wine (and to some extent, vinegar) are known to be attractive to some beetles. So, from the beginning, it was expected that sap beetles would be strongly represented in those traps that used beer and wine. To make this more applicable and generalizable, including some other of the commonly used solutions (e.g., antifreeze), would have been much appropriate.

Also, related to my previous comment, authors need to make it clear that trapping in their study is not all passive, given some of the solutions used (e.g., beer, wine or even vinegar) are known to be attractive to some species. Authors do use ‘bait’ and ‘unbaited traps’, both in the abstract and main document, and even in the keywords they include ‘attractive traps’ and ‘bait’ when referring to solutions, but no general context is provided. The use of baited vs unbaited traps has different purposes, which is not clearly defined in this manuscript, with baited traps purposely biased to collect those target organisms over others, which become by-catch. Therefore, I strongly recommend the authors to rewrite the abstract accordingly and make it clear in the introduction and methods, so the focus of the study is known to be directed towards attractants. This will provide a better focus on the purpose of the study, and will make statements, such as the closing sentence of the abstract, to be in context to the study objectives (as written, it was an unexpected and unconnected closing statement). I think authors need to explain further the 'biasing effect' they refer to on line 16 in the Abstract (and in the main text of the manuscript). In what terms the different solutions may bias the results? This is important and quite relevant as is the main focus of the study (based on the title).

Also, there is very little mentioned (none in the abstract) about the effect of exposure (shady vs. sunny) and its potential interaction with the different solutions in the capture of different assemblages of beetles. There is only one short statement (line 174) that refers to exposure in the results. Also, I don’t think the analytical approach to include both exposure and trap solution is appropriate. From what I can understand, this may have done through a two-way ANOVA (although it is not very clear). However, traps with different solutions are not independent from each other when considering exposure, therefore a mixed-model approach would be more appropriate to evaluating the effect of both factors and their interaction, with trap solutions nested within exposure levels (equivalent to a split-plot design).

The introduction, to me, is missing some general context, as described above, which will make it more directed to the objective of the study. I feel information is missing with respect to the purpose of trapping for biodiversity oriented studies, as well as to the inherent biases of trapping devices (regardless of the solution used to preserve the samples). Also, a bit more context on the solution biases would be useful, so readers would have a better understanding of the state of knowledge with respect to using different solutions. What authors present here is not novel, in the sense that it is known that some of the solutions used here do capture different species. So then, what is missing is what is the innovation from this study that makes is a contribution to what is already known with respect to trapping. With all respect to the authors, as is, the introduction seems more of a list of facts rather than a succession of ideas connected to each other that provide enough information to the reader to better understand the context of the problem and, as mentioned above, the novelty of the present study. I strongly recommend the authors to rewrite this section. I think the basic building blocks are there, but are not clearly described and connected.

Some information is missing form the ‘Data collection and sampling design’ section. For instance, the actual dates in which traps were active are missing. Also, and very importantly, there is no mention of what kind of traps was used for this study. There is a picture (Figure 2) shown but with no explanation. It is not clear form the description, if all traps were the same (I suspect they were) and how were they installed (where they all installed as shown in figure 2?, I hope so…). Some basic information of the size of the traps, more than just saying ‘a small bucket’ (line 94), is needed, the height at which they were installed (assuming all were all installed as in figure 2). I provide some suggestions below.

With respect to the analyses, the abundance and number of species are integers (count data) and therefore are not appropriate to be analyzed using gaussian models (those that assume a normal distribution). Count data are better assessed using generalized linear models with a correct underlying distribution (Poisson or Negative Binomial, if overdispersed). This approach is much better than having to transform the data to meet a normal distribution or to have to analyze data using both parametric and non-parametric methods. Also, see my comment above with respect to the experimental design of this study (i.e., split-plot). In relation to the multivariate analyses, I don’t really understand why there is a need to analyze the data twice using different (very different) approaches. One using constrained ordination (CCA) and one using unconstrained ordination (NMDS). To me, these two are redundant and if the objective is to look into beta diversity (as stated in line 134), CCA is a much more powerful approach.

Given the issues I identified above with respect to the analyses, I don’t provide much feedback on the results and discussion, as these will likely change.

I think the number of figures is quite large (17 figures) and thus, some can be grouped together. Please see some comments below with respect to the figures, as I think legends can be improved significantly and as well, need to be a more accurate description of what the figure(s) is(are).

Below are some specific suggestions and comments for authors to consider:

L17. ‘effectiveness’ in terms of what, number of individuals collected, number of species collected, diversity? Please specify. Also, please remove 'the most' as there are many other common solutions used.

L18. Please remove 'of the most studied group of insect'. Firstly, there are many other groups that are commonly collected that are also widely studied and secondly, beetles are insects, so in my opinion is ‘insect beetles’ is redundant.

L19. What type of traps was used as part of this study and for how long they were active? This is important information that is missing as there are many different types of traps. Also, it may read better ‘We placed six groups of four traps each (with either beer, wine, vinegar or water as preservative) in areas under two exposures (sunny and shady)’.

L20-22. Since the focus of the study is on beetles, I don’t think the total number of invertebrates collected is of any relevance. I’d suggest to rewrite this piece as something like ‘We collected a total of 3,931 beetles, of which 3,825 were identified into 120 species and 36 families’.

L22. It may be useful to provide how many species in Nitidulidae were collected too, and if there was a particular solution in which spa beetles were more commonly trapped. I bet, it was in traps with beer and wine

L23. There seems something to be missing connecting the abundance idea with the diversity piece. Would this suggested writing be what authors meant? ‘…very attractive and traps with these solutions collected similar beetle communities; however, the diversity (measured using Shannon’s index) was low due to the high…’

L24. Perhaps remove ‘On the other hand’ as I don’t really see the contrasting statement. Bear in mind that the solution is not what traps the beetles, is the actual trap. The solution is a preservative and killing agent, and perhaps an attractant, too in some cases (e.g., beer and wine). Therefore, ‘Traps with vinegar and water collected a similar composition and species richness’.

L26. ‘from all samples’. Also, I don’t really understand what authors mean with ‘between the single bait species’, is there perhaps a comma missing between ‘bait’ and ‘species’. Even then, the sentence is poorly structured. I strongly suggest authors to rewrite this for clarity.

L36. Please remove ‘much more’ and I am not sure if ‘frequent’ is the right word to use here. Also, it may be worth it for authors to expand better this idea and provide a much broader context, that connects better with the following sentence.

L37. Perhaps it is too blunt to say that beetles are ‘the most common studied group’. I’d suggest authors to tone down this and perhaps mention that beetles are one of the most common studied groups, instead.

L41. Perhaps ‘ocular’ is not the best wording for here?

L48. Word repetition. Can rewrite as ‘with different fixative solutions used to kill and preserve…’

L49-53. Remove ‘aquatic’. By definition a solution is aqueous and I don’t think is the right word use. This could be more effectively written as ‘Different solutions are commonly used in traps to collect invertebrates for biodiversity assessments, including brine…propylene glycol (C3H8O2; antifreeze)…copper sulfate (CuSO4), formaldehyde (CH2O)… acetic acid (CH3C00H; vinegar)’. Since authors provided the chemical formula for copper sulfate, for consistency, then it would be appropriate to include the formulas for the other compounds, which I have provided here; however, I think the formulas are perhaps not really needed as these are commonly known compounds. Will leave it to the authors to decide whether including them all or not.

L54. Fix typo on ‘coservative’, but even then I think ‘preserving’ is a better wording. I think here it would be a good place to provide background information on the use of beer and wine, given what the following sentence reads with respect to baits.

L74. ‘The reserve has an area’ to avoid word repetition.

L76. Is there any reference that can be cited with respect to the establishment of the protected area?

L77-78. There must be some reference to support the description of the main vegetation of the area. I would put a stop after ‘Hercynian oak woodlands’, followed by ‘The dominant tree species are…’, and remove the first ‘and’ as it is not needed. Is it possible to provide common names to the tree species? Also, not sure what the format of the journal is with respect to scientific names, but normally the author needs to be included as well. Please double-check.

L84. ‘Data collection’

L86. ‘Beetles were sampled between June 26 and July 30 2021 using small plastic buckets (provide dimensions and/or volume) with a roof suspended above to prevent rainfall (I assume) and debris into the trap. Traps were installed 30 m apart in groups of four at six locations (three on sunny and three on shady exposures) by attaching the bucket with wire to the bole of a tree at breast height (1.3 m). Traps in each group contained each one of four solutions: …. Samples were collected every four days… Beetles were sorted from the samples and identified to the family and species level (except for Aleocharinae (Staphylinidae), which were excluded (?). The remaining invertebrates in the samples were identified to the order level’

L118. ‘visualities’? I don’t understand what this refers to, but still, I don’t think this is the right wording.

L119. ‘species curves’? iNEXT is not meant to generate species curves. The analysis is meant to estimate species richness/diversity using a standardized reference for comparisons, in this case among trap solutions (and exposure conditions?).

L123. ‘Data were replicated’? I don’t think this is correct. Simply put, diversity was estimated using rarefaction (please confirm if coverage- or individual-based) and 95% confidence intervals obtained after 50 randomizations. Why 50 and not higher?


Figure 1. ‘…indicated in map’

Figure 2. The legend reads ‘trap types’ (plural), but only one image of a single trap is provided.

Figure 3 and figure 4 are confusing. I’m not sure if something is missing or misplaced. The legend reads ‘proportion of beetle abundance’; however, the plot shown shows abundances (not percentages) on the y-axis. Also, the legend refers to apparently two plots (‘invertebrate capture (left)’ and ‘invertebrate capture by bait medium (right)’), but only one plot is shown. Or is it that the ‘left’ plot is what is under Figure 3 and that the ‘right’ plot is under Figure 4? If that’s the case, then figure legends need to be adjusted.

I think the situation above applies as well for figures 6 and 7, and figures 8 and 9. Also I think Figures 10-13, Figures 14-15 and Figures 16-17 can be grouped together as a single figure

Experimental design

Please see my comments above

Validity of the findings

Please see my comments above

Additional comments

Please see my comments above

---

## Round 0.2 · Minor Revisions

This manuscript went through a previous round of review and due to a major revisions decision went out for a second round of review. The two reviews in the previous round covered a lot of important ground, and the authors have come a good distance in improving the paper.

Neither of the previous two reviewers was available for this round of review, so I found another expert reviewer who was graciously willing to review the manuscript. Rev. 3 had access to the previous reviews and the previous versions of the manuscript and has provided substantial and very useful comments. I agree with Rev. 3's comments, and while there are a fair number of items to revise, none of them rise to the level of a major revisions decision. As such, I don't anticipate that this manuscript will go out for another round of review, but instead, I will likely make a final decision on the paper once I have received a complete response and rebuttal.

Thank you to everyone who has taken part in this process -- reviewers and authors alike.

·

Excellent Review

This review has been rated excellent by staff (in the top 15% of reviews)
EDITOR COMMENT
This review is excellent because the reviewer took into account the results of two previous reviews in an earlier round and synthesized them and the revised manuscript in a way that made it clear where issues in the manuscript needed to be addressed. I particularly appreciated the care that the reviewer took to discuss different trapping fluids and why in some cases certain fluids that actually attract particular taxa may be useful compared to fluids with no behavioral activity. I believe this review will allow the authors to substantially improve their manuscript and increase its relevance to the scientific community if it receives final acceptance.

Basic reporting

The authors compared captures of insects in 1 litre bucket traps that vary in their trapping liquids (wine, beer, vinegar, and water) and exposure (i.e., sunny or shaded). The authors collected large numbers of arthropods but focused primarily on beetles, which in turn were comprised mainly of sap beetles (Nitidulidae). They found that trapping liquid had a significant effect on captures (abundance and species richness), with similar catches in beer and wine trapping liquids, and similar catches in vinegar/water trapping liquids. Once the sap beetles were removed (i.e., 86% of the total catch), catches across the four trapping liquids were more similar. The authors conclude that the choice of trapping liquid can lead to bias in the assessment of biodiversity due to the preferential attraction of certain groups to trapping liquids (i.e., sap beetles attracted to beer/wine).

There are many studies in the literature demonstrating potential biases of entomological traps (i.e., window traps, pitfall traps, etc.) related to characteristics of their design, preserving agent, and construction material. Despite the shortcoming of these inexpensive traps, they remain widely used and studies seeking to improve their efficiency or highlight potential biases, like the study in question, are needed. The authors have also used a trap design that is not-well studied in this regard (i.e., most studies focus on pitfall and window traps), and so the resulting beetle assemblage that is collected in these traps (i.e., predominance of Nitidulidae) also varies from most other studies and is worth reporting. However, the introduction (and also the title) are somewhat confusing in that they do not appropriately distinguish between the two common objectives of using ‘trapping liquids’. First, many trapping studies use a liquid killing agent and/or preservative with the intention of passively sampling the local environment. Potential bias may arise in this scenario if certain taxonomic groups are attracted to this liquid and such details are not factored into interpretation. Second, some studies focus on sampling particular taxonomic groups and make use of baits or attractants to increase captures of those groups (Lines 55-59). In this latter case, the taxonomic bias is assumed, and is usually intentional. I believe the authors understand this distinction (Lines 53-54), however, it is not clearly explained in the introduction and leaves the reader a bit confused when the trapping liquids chosen by the authors (e.g., beer and wine) are likely to serve as an attractant to certain groups. It is expected, for example, that choosing a trapping liquid that is an attractant (i.e., beer or wine) will ultimately increase captures of certain groups, and in such cases, a bias towards those groups is expected. As such, reporting that the catches in beer and wine are biased towards groups attracted to ethanol is not surprising.

The confusion surrounding the use of liquids as either attractants or preservatives was also noted previously by Reviewer 2 (paragraph 4 in Basic Reporting); however, Reviewer 1 suggested removing any discussion of preserving liquids given the authors used mainly attractants (Additional comments, paragraph 1). If the main focus of the study is to elucidate potential biases associated with trapping liquids, the authors should mention trapping liquids used as preservatives. Secondly, if the authors are intending to highlight problems associated with broader species richness estimates using baited traps (Lines 211-217), this should be more clearly articulated in the introduction.

The Tables and Figures used in the study are relevant and well-labeled with the following exceptions:
- ‘percentil’ in Figures 3 and 5 should be written percentile
- Figure 5 has four panels and could be labeled a, b, c, d. The two graphs on the right side of the figure have their Y-axis labeled "All_species" and "exceptNitidul_species" but it is unclear if this measurement refers to species richness. Given the authors use other diversity metrics in the paper (Shannon's diversity, rarefaction-based estimates, etc.)
- Figure 6 should read ‘Species accumulation curves’ instead of ‘Species cumulative curves’.

The raw data from the study are included in a table in the Supplemental material, but the data included in the excel file are in a strange format with many interspersed commas.

Experimental design

The experimental design is well suited to the study question (aside from perhaps using more preservative-type trapping liquids, see above) and the data support the authors conclusions. However, some details about the trapping liquids are missing and could be included. For example, the authors list “The following mixtures were arranged in traps: (i) pure red wine (1 litre + 1 kg sugar 1:1 and a spoonful of salt), (ii) beer (1 litre + 1 kg sugar 1:1 and a spoonful of salt), (iii) vinegar (8% acetic acid) and (iv) pure water as control”, but no volumes are reported for the vinegar or water. It is unclear, for example, if these mixtures were sub-sampled after the mixtures were combined. The authors report the volume of the traps to be 1 litre (Line 107), but it is unclear to me how 1 litre of wine plus a full kilogram of sugar can possibly fit in the trap. Also, were the trapping liquids replenished after each collection from the trap (Line 106)? The authors could also report the diameter of the opening of the trap as this information is especially useful in the context of evaluating trap biases and the broader trap characteristics that can influence captures (these data might be used in a meta-analysis, for example).

Validity of the findings

Results of the analysis support the main conclusions that choice of trapping liquid can bias catches.

Additional comments

1) The authors use ‘insolation’ to refer to traps placed in sunny or shaded areas. I suggest replacing the word ‘insolation’ with ‘exposure’ throughout the manuscript to more clearly emphasize that this variable refers to the degree of sun exposure for the traps.

2) The authors used indicator species analysis (line 158) to indicate attractiveness to certain trapping liquids, however, the authors also suspended the traps from different tree species. Although tree species was accounted for in their glmm (lines 131 and 132), it was likely not accounted for in the indicator species analysis. Thus, the effect of tree species may have influenced the resulting assessment in the indicator species analysis.


3) The random effect structure used in the authors’ generalized linear mixed effects model is a bit confusing (Lines 130-132). Typically, random effects are used for variables that are influential but of less interest to the study question, however, the authors report p-values and chi-square values for 'insolation' in Table 1. The authors also used a random effect on the trapping liquid (lines 131-132), which is confusing considering this was related to their original study question. The authors should provide an explanation in the methods for why they used the random effect structure in this way. Alternatively, the authors could use trapping liquid and exposure/insolation as fixed effects (with hanging tree species as a random effect), and explore potential interactions between exposure and trapping liquids. Beer and wine trapping liquids exposed to greater solar radiation, for example, might attract higher numbers of arthropods.

4) The study is missing other important references related to testing biases in preservative liquid among traps including Luff 1968 and Koivula et al. 2003. See also the 'Choice of Killing Preservative' section in the recent review by Brown and Matthews 2016. These studies examined similar questions, albeit, in the pitfall traps. References to these papers are as follows:

Luff, M. L. 1968. Some effects of formalin on the numbers of Coleoptera caught in pitfall traps. Entomol. Mon. Mag. 104: 1247 - 1249

Koivula, M., D. J. Kotze, L. Hiisivuori, and H. Rita. 2003. Pitfall trap efficiency: do trap size, collecting fluid and vegetation structure matter? Entomol. Fenn. 14: 1 - 14.

Brown, G.R., I.M. Matthews. 2016. A review of extensive variation in the design of pitfall traps and a proposal for a standard pitfall trap design for monitoring ground‐active arthropod biodiversity. Ecol Evol. 6: 3953–3964.

5) Line 193 - authors note 'species richness curves' but it may be more appropriate to say 'species accumulation curves'. Also, worth specifying that “wine had the highest observed diversity” … not "wine has the highest species richness"), and that by saying "and beer has the highest attraction" the authors mean in terms of total individuals collected. As written, the language is a bit confusing and unclear.

6) Lines 195 - Shannon diversity 'showed' not 'shows'. Lines 197 - 'differed' not 'differ' to keep the tense consistent.


7) Lines 196 - Wording could be changed to "Nevertheless, after removal of Nitidulidae, these differences were not significant".

---

## Round 0.3 · accepted · Accept

Following several rounds of review by three referees, and substantial and meaningful work by the co-authors to improve their paper, this manuscript is now ready for publication in PeerJ. In particular, the co-authors have now done a better job of contextualizing their work in terms of current practice, and the results should help to guide beetle sampling methods.

Thank you once again for the work done by the reviewers and co-authors during this process.